# Relative changes in brain and kidney biomarkers with Exertional Heat Illness during a cool weather marathon

**Michael J. Stacey**[1,2,3]*, **Neil E. Hill**[4], **Iain T. Parsons**[1,3,5], **Jenny Wallace**[6], **Natalie Taylor**[6], **Rachael Grimaldi**[7], **Nishma Shah**[8], **Anna Marshall**[8], **Carol House**[9], **John P. O'Hara**[3], **Stephen J. Brett**[2,10], **David R. Woods**[1,3]

**1** Academic Department of Military Medicine, Royal Centre for Defence Medicine, Birmingham, United Kingdom, **2** Department of Surgery and Cancer, Imperial College London, London, United Kingdom, **3** Carnegie School of Sport, Leeds Beckett University, Leeds, United Kingdom, **4** Department of Metabolism, Digestion and Reproduction, Imperial College London, London, United Kingdom, **5** School of Cardiovascular Medicine and Sciences, King's College London, London, United Kingdom, **6** National Health Service (Wales), United Kingdom, **7** Brighton and Sussex University Hospitals NHS Trust, Brighton, United Kingdom, **8** University College London, London, United Kingdom, **9** Environmental Medicine Services, Institute of Naval Medicine, Gosport, United Kingdom, **10** General Intensive Care Unit, Hammersmith Hospital, London, United Kingdom

* m.stacey13@imperial.ac.uk

**Data Availability Statement:** All relevant data are within the manuscript and its Supporting information files.

## Abstract

### Background

Medical personnel may find it challenging to distinguish severe Exertional Heat Illness (EHI), with attendant risks of organ-injury and longer-term sequalae, from lesser forms of incapacity associated with strenuous physical exertion. Early evidence for injury at point-of-incapacity could aid the development and application of targeted interventions to improve outcomes. We aimed to investigate whether biomarker surrogates for end-organ damage sampled at point-of-care (POC) could discriminate EHI versus successful marathon performance.

### Methods

Eight runners diagnosed as EHI cases upon reception to medical treatment facilities and 30 successful finishers of the same cool weather marathon (ambient temperature 8 rising to 12 ºC) were recruited. Emerging clinical markers associated with injury affecting the brain (neuron specific enolase, NSE; S100 calcium-binding protein B, S100β) and renal system (cystatin C, cysC; kidney-injury molecule-1, KIM-1; neutrophil gelatinase-associated lipocalin, NGAL), plus copeptin as a surrogate for fluid-regulatory stress, were sampled in blood upon marathon collapse/completion, as well as beforehand at rest (successful finishers only).

### Results

Versus successful finishers, EHI showed significantly higher NSE (10.33 [6.37, 20.00] vs. 3.17 [2.71, 3.92] ug.L$^{-1}$, P<0.0001), cysC (1.48 [1.10, 1.67] vs. 1.10 [0.95, 1.21] mg.L$^{-1}$, P =

**Funding:** The sources of funding for the study were (a) direct financial support from the Ministry of Defence to cover assay costs, salaries of involved military personnel, their temporary accommodation if required close to the study site and (b) material hosting by the Brighton marathon medical team, who provided medical tentage in which the study was performed and appropriate screens to protect volunteer dignity while being sampled etc. No formal numbered grant award was made, rather the MoD raised and settled purchase orders e.g. with Affinity Biomarkers, the commercial company that assayed the chemistry reported. The funders had no role in study design, data collection and analysis, decision to publish, or preparation of the manuscript.

**Competing interests:** The authors have declared that no competing interests exist.

0.0092) and copeptin (339.4 [77.0, 943] vs. 18.7 [7.1, 67.9] pmol.L$^{-1}$, P = 0.0050). Discrimination of EHI by ROC (Area-Under-the-Curve) showed performance that was outstanding for NSE (0.97, P<0.0001) and excellent for copeptin (AUC = 0.83, P = 0.0066).

## Conclusions

As novel biomarker candidates for EHI outcomes in cool-weather endurance exercise, early elevations in NSE and copeptin provided sufficient discrimination to suggest utility at point-of-incapacity. Further investigation is warranted in patients exposed to greater thermal insult, followed up over a more extended period.

## Introduction

As global temperatures rise, elite athletes, sports participants, recreational runners and hikers, and a range of occupational groups share an increasing risk of EHI (Exertional Heat Illness). A spectrum of EHI [1–3] severity is described in the literature, ranging from transient reductions in consciousness and lesser elevations in core temperature with mild EHI to the pronounced hyperthermia, central nervous system (CNS) dysfunction and multi-organ failure observed in exertional heat stroke (EHS) [4]. In civilian mass participation events, the occurrence of severe EHI may reach 31 case per 10000 competitors in higher-risk races [5]. A lower EHS incidence of 1 in 10,000 finishers has been reported in marathon runners [6], though cases may increase with the severity of climatic conditions [7] and can be even greater in and other athletic disciplines [8].

Larger scale mass participation recreational events may attract clinical and laboratory support familiar with the presentation and management of EHI. Capacity and capability may be more limited, however, for detached emergency responders, 'pop-up' medical facilities at smaller events and in certain industrial and agricultural settings. This risks EHI events being under-estimated, under-resourced or under-managed by non-specialists. Indeed, where prior experience is lacking and EHI not recognised or distinguished from benign exertional hyperthermia, cooling may be altogether neglected [4]. This is highly relevant to more remote field settings, including those with a military focus, in which the need for early recognition of end-organ injury—for example acute kidney injury (AKI)—has been highlighted [9]. In such circumstances, prompt decisions regarding escalation and evacuation to higher echelons of medical care may have a definitive impact on outcome both for the individual and the event or mission [4].

On occasion, EHS may manifest in an indolent or subacute fashion, whereby initial clinical progress provides false reassurance [10]. Even with fulminating illness, standard laboratory parameters may not immediately or reliably reflect or predict the risk of organ injury [4, 11]. As such, the safety and appropriateness of releasing from medical care or advising on return-to-exercise can be difficult to judge in a patient affected by EHI, especially in the relatively short-space of time available to emergency providers. This may be particularly important where the affected individual will be unaccompanied upon discharge, or subsequent heat tolerance testing is available and relatively indicated [12]. In such circumstances, an early indication of established or evolving organ injury could help inform whether ongoing clinical supervision is indicated and/or aid assessment of the likelihood of increased risk associated with further bouts of exercise and thermal stress [13]. In this sense, a reproducible

physiological index to discriminate EHS with downstream clinical sequelae from lesser kinds of incapacity could help tailor a clinical pathway from collapse to recovery.

We have demonstrated compromise to the GI epithelium with early, substantial and sustained elevations in intestinal fatty acid binding protein (I-FABP) in marathon runners diagnosed with EHI [14]. This raises the question of whether these patients experienced significant injury to other organ systems in the evolution of their illness, which might have been detectable at the initial point of care (POC). As central nervous system (CNS) impairment was defining and renal stress anticipated with the substantial cardiovascular strain observed, we further investigated biomarkers with emerging clinical utility in relation to brain (neuron specific enolase, NSE; S100 calcium-binding protein B, S100β), kidneys (cystatin C, cysC; kidney-injury molecule-1, KIM-1; neutrophil gelatinase-associated lipocalin, NGAL) and fluid-regulatory stress (copeptin, the C-terminal part of pro-arginine vasopressin that is a stable surrogate for arginine vasopressin, AVP).

This selection of biomarkers was informed by previous investigations in humans showing prognostic potential when measured close to the point of injury/debility, as for heat stroke (S100B) [15], traumatic and septic encephalopathy (NSE) [16, 17] and clinically significant AKI of various aetiologies (sCr, NGAL, KIM-1) [18, 19]; as well as by studies showing associations with inflammatory responses specific to acute exercise-heat stress (NGAL) [20] and subclinical AKI (copeptin) [21, 22]. Our primary aim was to determine whether these markers could effectively discriminate marathon runners affected by EHI from successful finishers completing the event that same day. Secondary aims were to examine changes in these biomarkers over time and to relate indirect evidence for organ injury to likely precipitants, co-factors or other indicators relevant to the episodes of collapse observed.

## Materials and methods

We collected data from two groups of runners–people who volunteered for blood and anthropometric assessment to occur before and after the marathon (henceforth referred to as 'successful finishers') and a separate group of people not enrolled as a successful finisher, who collapsed and required medical treatment for EHI during the marathon. No successful finishers collapsed or required medical treatment during the marathon event.

### Successful finishers

Potential participants were contacted via electronic mailshot from the Brighton Marathon Race team and provided with information pertaining to associated research projects. Inclusion criteria were age 18–60 years. Volunteers were excluded if they had not read and complied with the Medical Advice for participating in the marathon (http://www.brightonmarathonweekend.co.uk/medical-advice/), had a prior history of heat illness, significant preceding head injury, epilepsy, congenital or acquired kidney disease or were taking drugs known to affect the renal system (including non-steroidal anti-inflammatory drugs).

Anthropometric and physiological measurements and blood tests were taken between 10.00 and 19.00 on the day prior to the event. Unshod standing height and minimally clothed body mass were recorded for each control participant using a stadiometer and scales. Participants were then seated for around 10 min prior to the measurements of resting heart rate (HR), systolic (SBP) and diastolic blood pressure (DBP) all using an integrated patient monitoring device (GE Carescape V100, UK). Venepuncture was performed at the antecubital fossa.

Blood samples (13 ml per draw) for these successful finishers were taken at the following time-points: (1) 'pre-race' baseline (B), at race registration the day before the marathon; (2)

T0; as close to the time of successful completion of event as feasible (<30 minutes); (3) T24; the following day, as close to 24 hours post-run as feasible. Blood was centrifuged at 1500G for 15 minutes then separated and snap frozen in liquid nitrogen on site for subsequent assay.

## Collapsed runners

Runners who collapsed during the marathon were assessed for study enrolment following evacuation to the nearest marathon medical facility, clinical re-assessment and immediate essential medical treatment. The cases in question were triaged, treated and confirmed by clinicians experienced in the management of EHI, who were able to discount non-EHI diagnoses including post-exertional hypotension, primary cardiac disorders and exercise-associated hyponatraemia uncomplicated by hyperthermia. Criteria for recruitment were a clinical diagnosis of EHI, where excess body heat was deemed the primary cause of incapacity and core body temperature measured rectally (Intellivue integrated thermistor, Philips Healthcare, Amsterdam, Netherlands) was ≥38.5°C, in association with CNS impairment (for example, abnormal motor control, loss of responsiveness, amnesia for the episode) occurring spontaneously during or soon after marathon run followed by failure to make a prompt recovery with prostration and initial medical care. Explicitly, runners diagnosed with exertional or post-exertion syncope were excluded from recruitment, as were those re-categorised with an alternative aetiology based upon response to initial treatment. Level of consciousness upon presentation to medical staff was defined according to the widely used Alert-Voice-Pain-Unresponsive (AVPU) scale, which assigns the best casualty response to stimulation in a graded fashion and corresponds with the more detailed Glasgow Coma Scale as: <A> Alert: awake but potentially confused, GCS up to 15; <V> Verbal: responsive to verbal stimulation, GCS ~12; <P> Pain: responsive to painful stimuli, GCS ~8; <U> Unresponsive: unconscious, no response to voice or pain, GCS 3.

Blood samples (13 ml per draw) were taken at the following time-points: (1) T0; as close to the time of collapse as feasible (within 30 minutes), (2) T1; at 1 hour following collapse (as clinical considerations allowed) and (3) T4; at 4 hours following collapse (again as clinical considerations allowed). No EHI cases were available for next-day (T24) sampling despite ethical approval being in place to do so. Samples were analysed at Affinity Biomarker Labs (London, UK). Serum was analysed for sCr, cysC, CK and total protein (TP) on a commercial platform (Siemens Advia 1800, Siemens Healthcare Diagnostics Ltd, Camberley, UK). Serum NSE, KIM-1 and NGAL (R & D Systems Europe, Abingdon, UK) were measured by commercially available immunoassay with intra- and inter-assay variability of <10% and an upper limit of detection of $20ug.L^{-1}$, $700\ ng.L^{-1}$ and $200\ ug.L^{-1}$ respectively. Serum S100β (EMD Merck-Millipore, St. Louis, USA) was measured by sandwich ELISA with variability <5% and upper limit of detection of $200\ ng.L^{-1}$. Plasma copeptin was measured with Time-Resolved Amplified Cryptate Emission technology (Thermo Fisher-Brahms, Hennigsdorf, Germany).

Data were assessed for normality and expressed as mean ± SD or median [IQR]. Linear relationships between parametric and non-parametric variables were assessed for significance by Pearson's or Spearman's rank tests, respectively. Data were compared by t-test (parametric data) or Mann-Whitney test (non-parametric data) and Receiver Operating Characteristic (ROC) curves were constructed to determine the Area-Under-The-Curve and sensitivity and specificity for the biomarkers of interest in discriminating EHI versus successful finishers.

Significance was set to alpha = 0.05. A formal power calculation was not attempted prospectively, due to the study protocol's dependence upon a convenience sample of willing volunteers and uncertain availability of EHI cases. However, it was noted that, in a sample of 28 patients affected by brain injury with a traumatic mechanism, power was adequate to discriminate 9

survivors with favourable functional outcomes from 19 patients experiencing death or poor functional outcomes according to serum NSE and S100β measured soon after insult, with peak sensitivity and specificity of 88% and 100% for S100β [23]. Elsewhere, a study of 10 subjects showed significantly greater NGAL with exercise under heat stress where increased inflammation and elevated kidney injury biomarkers were induced versus control conditions in the same subjects [20]. With marathon running, both plasma copeptin and NGAL were shown to be significantly higher in 12 runners with AKI defined biochemically versus 10 experiencing sub-threshold elevation in sCr [21].

### Ethics approval

Ethical approval for a study of people running the 2019 Brighton Marathon was obtained from London South East ethics committee (19/LO/0340 247967). At the race registration (on the day before the marathon) volunteers (successful finishers) were formally recruited to take part in the study and provided written informed consent. For EHI cases who initially lacked mental capacity to consent for themselves (AVPU grade V to U, or A with any concern for capacity) we proceeded with presumed consent until they were deemed able to give it retrospectively.

## Results

On the day of the marathon, ambient temperature measured at the local meteorological station increased from 8 ˚C during the event muster (race start time 09:45) to peak at 12 ˚C with runners still on the course between 14:00 and 15:00. Among runners who collapsed while participating and underwent triage upon reception to medical facilities, eight underwent serial clinical review and were confirmed as EHI cases. These runners (5 male, 3 female) had completed a maximum of 22.4 ± 5.5 miles before collapsing and being evacuated forwards to the nearest on-course medical facilities, stationed at, respectively, 14 and 26.3 mile points. The recruited cohort represented 100% of clinically confirmed EHI cases presenting to marathon medical staff, at the only hospital-standard receiving facilities available on-course.

Individual biochemical results and available clinical observations are displayed in Table 1. In the global pandemic context, platform availability and reagent shortages limited measurements of copeptin to 7 of these 8 cases. S1 File shows comparative further biochemical measurements obtained 1 or 4 hours post-collapse, with NSE seen to decline in 2 out of 3 cases with this more complete data.

Thirty healthy runners aged 37.7 ± 8.9 years old (16 male, 14 female) successfully completed the marathon distance of 26.2 miles in 4.2 ± 0.8 (range 2.8 to 5.6) hours. Loss of body mass was -2.5 ± 1.6%. S2 File displays physical and biochemical results from B and T0, including laboratory reference ranges in health. For runners who also completed T24 measures (n = 18), sequential changes B-T0-T24 are displayed in Fig 1 and S3 File. For T0 measures, NGAL correlated with T0 sCr (r = 0.63, p = 0.005); S100β correlated with T24 CK (r = 0.66, P = 0.0028); and CK correlated with T24 KIM-1 (r = 0.55, p = 0.018).

Biochemical profiles for EHI cases versus successful finishers are shown in Fig 2. Analysis by ROC with corresponding AUC, where significant, and performance values (EHI versus successful finishers) is displayed in Fig 3. Versus successful finishers, EHI cases had higher heart rate (131 ± 20 vs 87 ± 14 b.min⁻¹) and lower diastolic blood pressure (58 ± 9 vs 67 ± 8 mmHg, P = 0.0123), but showed no difference (P = 0.4677) in systolic blood pressure.

## Discussion

This study is the first to examine a number of blood biomarkers for human EHI, both in heat-attributed collapse and with successful marathon completion. Our investigation also

**Table 1. Clinical and biochemical results for eight EHI cases sampled within 30 minutes of incapacity (T0 sampling point).** Loc. 1. Treatment facility stationed at 14 mile-point on course (NB course design resulted in runners up to 21 miles being received here). Location II. Main medical tent stationed 100 m behind finishing line.

| Loc. | Case/ Time | Distance | AVPU | Tc °C | RR breaths. min$^{-1}$ | HR beats. min$^{-1}$ | BP mmHg | NSE ng. L$^{-1}$ | S100b ng.L$^{-1}$ | sCr μmol.L$^{-1}$ | cysC mg.L$^{-1}$ | NGAL ug.L-1 | KIM-1 ng.L$^{-1}$ | Copeptin pmol.L$^{-1}$ | CK IU. L-1 | TP g. L-1 |
|---|---|---|---|---|---|---|---|---|---|---|---|---|---|---|---|---|
| I. | 1 | 14 miles | V | 40.1 | 60 | 115 | 101/46 | 6.37 | 90.52 | 154 | 1.53 | 185.48 | 14.04 | 339.4 | 179.0 | 78.2 |
| | 2 | 25 miles | A | 41.5 | 25 | 155 | 117/57 | 6.37 | 184.79 | 155 | 1.74 | >200.0 | 27.74 | U/A | 324.6 | 77.3 |
| | 3 | 21 miles | A | 39.0 | 50 | 136 | 120/52 | >20.00 | 98.97 | 145 | 0.94 | 112.94 | 52.56 | 104.0 | 305.7 | 74.3 |
| II. | 4 | >21 miles | V | 39.1 | 40 | 106 | 138/68 | 9.9 | 111.91 | 103 | 1.08 | 98.52 | 22.40 | 983.3 | 310.6 | 63.8 |
| | 5 | >21 miles | P | >40.0 | U/A | U/A | U/A | 26.78 | 38.63 | 124 | 1.16 | 136.12 | 30.00 | 77.0 | 729.5 | 56.3 |
| | 6 | >21 miles | V | 38.6 | 28 | 141 | 109/65 | 4.29 | 232.06 | 192 | 1.77 | 351.2 | 19.31 | 942.6 | 484.9 | 75.6 |
| | 7 | >21 miles | V | >38.5 | U/A | U/A | U/A | >20.00 | 98.33 | 142 | 1.48 | 179.44 | 26.69 | 5.6 | 802.7 | 67.1 |
| | 8 | >21 miles | V | >38.5 | U/A | U/A | U/A | 10.76 | <2.7 | 164 | 1.59 | >200.0 | 40.61 | 802.7 | 603.0 | 68.7 |

AVPU, Alert-Voice-Pain-Unresponsive scale; BP–blood pressure; CK–creatine kinase; cysC–cystatin C; HR–heart rate; KIM-1 –Kidney Injury Molecule 1; NGAL–neutrophil gelatinase associated lipocalin; NSE–neuron specific enolase; RR–respiratory rate; sCr–serum creatinine; Tc -first measured core temperature upon admission to medical facility ± 1 hour post-admission Tc; TP–Total protein; U/A–data unavailable (data entry spoiled, however all entered participants known to have Tc measured upon reception to medical facility >38.5 ˚C).

represents a relatively unique examination of EHI generated against the environmental gradient by metabolic heat production, in runners who faced challenging conditions of relatively cold Spring temperatures and gusting headwinds along the final seafront miles of the marathon course. Despite a tendency for on-course cooling secondary to these conditions, novel variation in NSE, cysC and copeptin was evident between clinically-diagnosed EHI cases and successful finishers. The striking elevations in NSE and copeptin observed in some collapsed runners suggested a substantial degree of neuronal as well as fluid-regulatory and/or cardio-vascular stress at point of collapse. This associated with case discrimination by ROC that was outstanding for NSE (AUC>0.9), excellent for sCr and copeptin (>0.8) and robust for cysC (>0.7).

Serum NSE is a neuronal enzyme indicative of neuroinflammation [17], which has been reported to rise early in sepsis-associated encephalopathy (SAE) and show positive associations with elements of systemic inflammation, including IL-6 [18]. Therefore NSE might be expected to increase with the evolution of endotoxaemic encephalopathy concomitant to exertional hyperthermia. This is suggested by the present work and supported by elevated I-FABP in our complementary reporting of the same casualties [14]. While the level of consciousness observed upon initial medical assessment was less depressed in these EHI cases than observed in other positive clinical studies of patients with SAE [18], traumatic brain injury [23] and cardiac arrest [24], this tallies with a discriminant value of NSE in the present investigation that was relatively lower in numerical value. This would also be in keeping with lower grade neuronal injury sustained with evolving EHI, thankfully interruptible with cooling–first passively upon cessation of exercise, then actively under medical supervision–and fortunately associated with eventual recovery to baseline in all cases.

Unlike NSE, S100β was not elevated in EHI cases versus successful finishers. This accords with its lack of discrimination for encephalopathy and lower prognostic potential when measured in the early stages of injury among other cohorts. As a marker of functionally milder

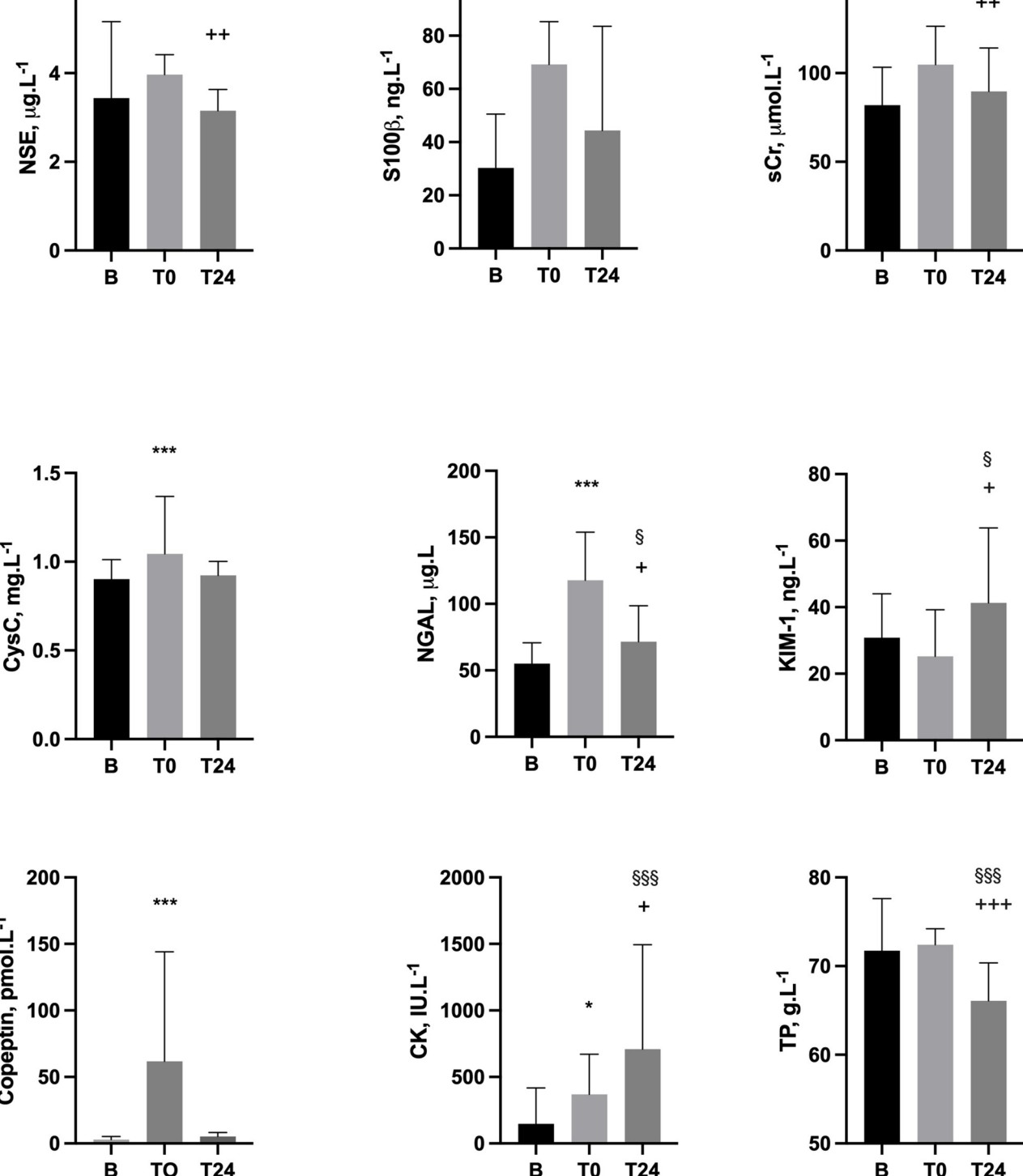

**Fig 1. Biochemical variables assayed pre-marathon (B), upon completion (T0) and next-day (T24) in 18 successful finishers with complete data across the three measurement points.** T0 vs B ***P<0.0005, **P<0.005, *P<0.05. T24 vs B §§§P<0.0005, §§P<0.005, §<0.05. T24 vs T0 +++P<0.0005, ++P<0.005, +P<0.05. *CK–creatine kinase; cysC–cystatin C; KIM-1 –Kidney Injury Molecule 1; NGAL–neutrophil gelatinase associated lipocalin; NSE–neuron specific enolase; sCr–serum creatinine; TP–Total protein.*

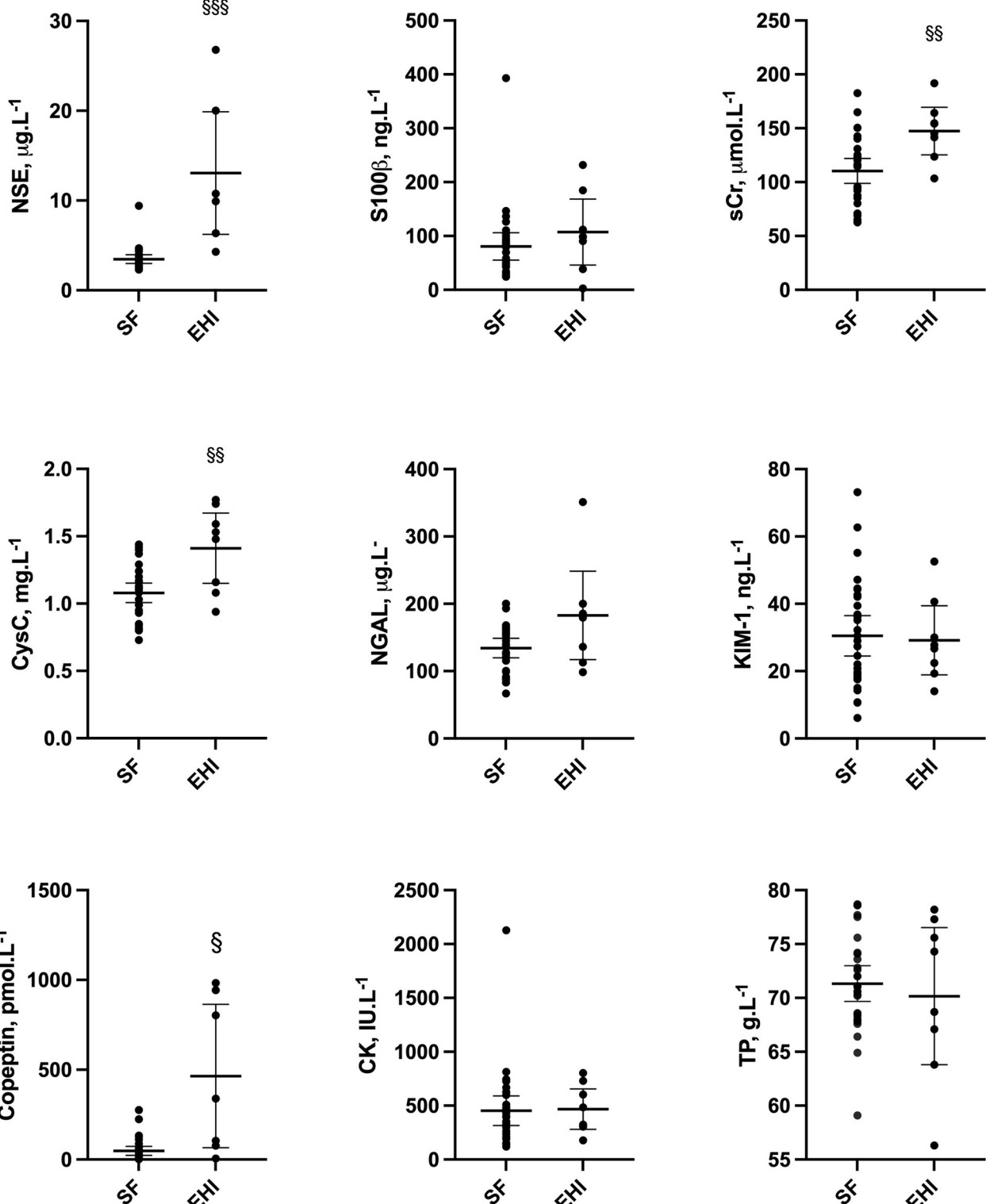

**Fig 2. Biochemical comparison at T0 (marathon completion or withdrawal) for 30 successful finishers (SF) vs 8 Exertional Heat Illness cases (EHI)\*** §§§P<0.0005, §§P<0.005, §<0.05. *n = 7 for copeptin. *CK–creatine kinase; cysC–cystatin C; KIM-1 –Kidney Injury Molecule 1; NGAL–neutrophil gelatinase associated lipocalin; NSE–neuron specific enolase; sCr–serum creatinine; TP–Total protein.*

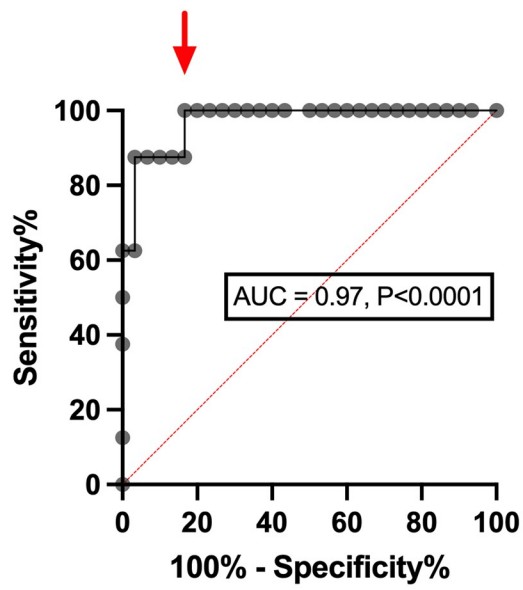

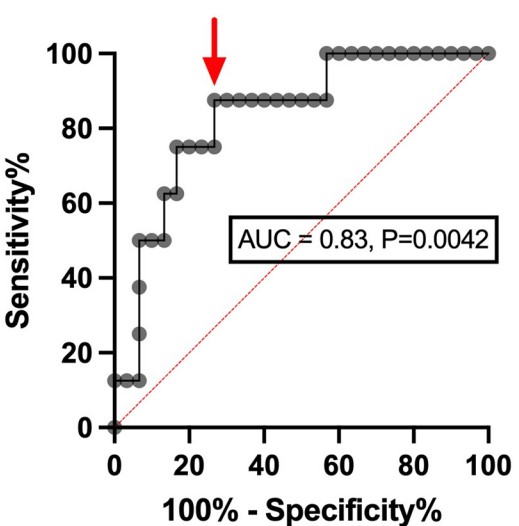

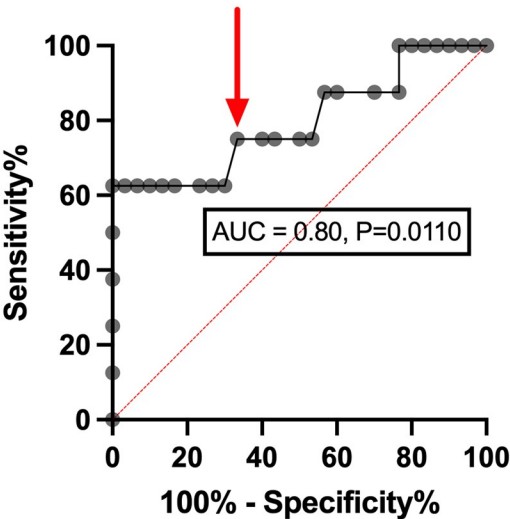

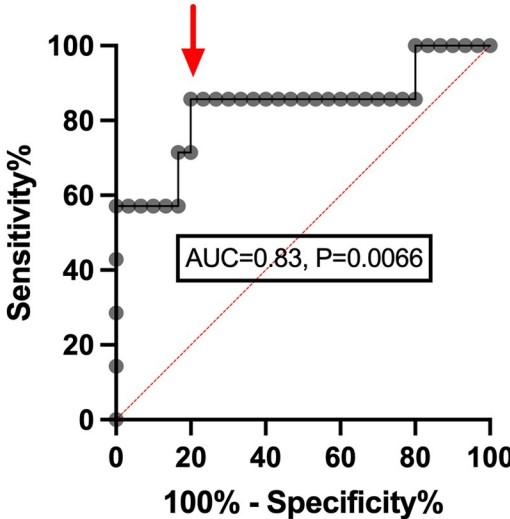

**Fig 3. ROC curves for biochemical parameters showing Area-Under-Curve (AUC) ≥0.7 for 30 successful finishers (SF) vs 8 Exertional Heat Illness cases (EHI),** [*] **assayed at T0 (marathon completion or withdrawal).** *CysC–cystatin C; NGAL–neutrophil gelatinase associated lipocalin; NSE–neuron specific enolase; sCr–serum creatinine.* [*]n = 7 for copeptin.

brain insults associated with lesser reductions in consciousness at presentation (Glasgow Coma Scale score 13–15, equivalent to 'A' or 'V' on the AVPU scale), S100β has reported high sensitivity for radiographically-demonstrable traumatic intracranial injury, at a threshold corresponding to the mean and median values observed in EHI cases in the present work (~100ng.L$^{-1}$) [24]. The specificity of S100β for brain injury is reduced in trauma encompassing extracranial sites, however, perhaps due to release from other tissues including those containing chondrocytes and fat cells [25]. In the present work, correlation with next-day CK suggest that S100β measured close to the point of collapse may have additionally reflected musculo-skeletal stress in successful finishers. Being renally excreted with a half-life of 30 to 120 minutes, S100β would be also be expected to show an effect of exercise-associated changes in GFR and AKI in addition to any attributable to any CNS injury.

Risks of dehydration in this study are likely to have been lower than many comparable reports in the literature. Among successful finishers, this was evidenced by loss of body mass that was not excessive for endurance exercise and lower copeptin than has been observed in warmer marathon conditions [21], perhaps due to relatively reduced osmotic/volume stimulus to AVP secretion. In EHI, a range of copeptin values were observed, with 6 out of 7 cases assayed showing levels more than twice the median of successful finishers; several values reached or exceeded those reported in severe sepsis and haemorrhagic shock [26, 27]. This may indicate relatively greater dehydration altogether, with both hypovolaemia and hypertonicity being potent, classical stimuli to the release of AVP/copeptin. Incipient inflammation, cardiovascular stress, reduced renal clearance and AKI or hyperthermia may also have contributed to elevated copeptin levels [22]. In one of two EHI cases sampled 1 hour post-recruitment to the present study, relative doubling of copeptin and KIM-1 suggested renal injury from ongoing relative ischaemia, despite cooling.

Taken together, these findings are supportive of further enquiry into the utility of a number of the biomarkers described in identifying and prognosticating for severe EHI. The requirement for such a tool may be debated, as the majority of EHI patients treated promptly with cooling show a good prognosis following incapacitation [4]. However the perennial challenge of managing heat stress derived from both exercise and the environment, repeated instances of fatal EHS where preventive guidelines were in place (if not wholly followed) and potential difficulties in consistently recognising and differentiating significant EHI from lesser forms of incapacity, especially in cooler conditions, indicate a need for better tools for attendant health-care professionals. The American College of Sports Medicine acknowledges in its position statement that the clinical changes associated with EHS can be subtle and easy to miss if coaches, medical personnel, and athletes do not maintain a high level of awareness [28].

Athletes deemed to have suffered moderate to severe EHI may be advised to follow-up with a physician after initial recovery and discharge from local medical facilities [12, 28]. Practically speaking, this extends the importance of early, accurate diagnosis into considerations regarding return-to-play/participation. Whether some of the cases reported above represented EHS or lesser forms of EHI may be rather academic in real-time; what may matter more is the ability to detect those at risk of organ-injury early on, to ensure that appropriate management is instituted and followed through to sporting, recreational or occupational recovery. This first requires research tools to help stratify casualties for interventions auxiliary to cooling and determine how initial presentation and treatment relate to longer-term outcomes [9].

One potential limitation of the study is that emergency care providers around the course may have elected to transfer potential EHI cases with more concerning physiology direct to the local permanent hospital Emergency Department, rather than delivering them to the on-course medical facilities from which we recruited. However the triage of marathon casualties is co-ordinated by radio communication with a central hub of medical advice co-located with

the main temporary (finish line) medical facility and we were made aware of no other severe cases bypassing this facility, nor did Emergency Department staff at the local hospital report other casualties attending when our team attended in follow up of case 5 at 4 hours post-collapse. Some unidentified EHI cases may have recovered ambulatory capacity and failed to attend for any medical treatment on-course or to present to hospital facilities, though these would almost certainly have represented the milder end of the spectrum of EHI presentations described in the literature, rather than being heat stroke cases.

We acknowledge that in recruiting cases close to the point of individual collapse along the marathon course, the final running distance completed varied within the EHI cohort. A disparity was also present in comparison to successful finishers at T0, who had all completed the full 26.2 mile run. Nevertheless, sampling occurred within a standardised time window (<30 minutes of EHI or marathon completion) and resulted in biomarker values that were largely more pronounced, rather than less, in the EHI group.

Co-factors in the evolution of EHI may also have impacted results in the direction of increased biomarker values, e.g. haemolysis arising with hyperthermia, which may represent an early stage in the development of clotting abnormalities in heatstroke, but can also increase NSE release from erythrocytes [29]. Another limitation is that the exclusion criteria for the successful finisher cohort (listed in the Methods) mean that this group may differ from the collapsed athlete group, who were not subject to these exclusion criteria. Whether collapsed EHI cases run faster, generate more metabolic heat are relatively more or less endurance trained than successful finishers in this context are all relevant questions, reflecting important mechanistic considerations.

However shortfalls in data of this kind need not detract from the practical potential reported in the present work and future studies may be able to recruit, follow and characterise sufficient collapsed cases to be more stringent in this respect. Better demographic characterisation (age, anthropometric characteristics) and improved understanding of the effect of changes in plasma volume on these markers (e.g. Dill and Costill-corrected values [30] for paired data, as in successful finishers) would, respectively, provide reassurance on the generalisability of these initial findings and relevant mechanisms underpinning them. This will require increased resourcing and logistical support to more completely overcome the unfavourable conditions for uniform collection of contextual data in this kind of field setting.

The investigation of a wider panel of biomarkers of relevance to renal and brain injuries may also be considered. For example, glial fibrillary acid (GFAP) has shown prognostic potential in detecting radiographically occult traumatic brain injury at presentation [31]. Although not included in this preliminary investigation, due to lack of data supporting associations with other relevant forms of encephalopathy at the time of writing, GFAP has since shown potential to assist early diagnosis and prognostication for sepsis-associated encephalopathy [32]. This comes with the added possibility of POC measurement [33], which could be highly advantageous in the marathon medical environment, specifically in relation to a diagnosis of heat stroke whereby systemic inflammation shares many parallels with sepsis and as such may similarly benefit from early intervention [4].

In conclusion, both emerging biomarkers in the form of NSE and copeptin, and more established injury surrogates such as sCr, performed well under cold weather conditions in discriminating successful marathon performance from supervening EHI. With significant renal injury identified in 16–25% of individuals affected by heat stroke of both kinds [2–4], more work is clearly required in this area. Later measurement of both neuro- and renal biomarkers (e.g. Day 1 and Day 3 post-event) could build prognostic value into these insights from the initial hours following collapse and allow further scrutiny for differences versus successful finishers. Extending this enquiry into other thermally and/or physically stressful

settings may have potential utility for a range of sporting, recreational and occupational activities, especially where an episode of incapacity needs further substantiation to inform clinical decisions on rest, recuperation and 'return to play/work'.

## Supporting information

**S1 File. Clinical and biochemical results for three EHI cases sampled following collapse, within 30 minutes of incapacity (T0) and again 60 minutes (T1) or 4 hours (T4).** Loc. 1. Treatment facility stationed at 14 mile-point on course (NB course design resulted in runners up to 21 miles being received here). Location II. Main medical tent stationed 100 m behind finishing line.
(DOCX)

**S2 File. Results for 30 successful finishers, at rested baseline B and upon marathon completion T0.** CK–creatine kinase; cysC–cystatin C; KIM-1 –Kidney Injury Molecule 1; NGAL–neutrophil gelatinase associated lipocalin; NSE–neuron specific enolase; sCr–serum creatinine; sNa–serum sodium; Ur–serum urea; TP–Total protein.
(DOCX)

**S3 File. Individual biochemical results for 18 successful finishers (n = 15 for copeptin), at rested baseline B, upon marathon completion T0 and next-day T24.** *CK–creatine kinase; cysC–cystatin C; KIM-1 –Kidney Injury Molecule 1; NGAL–neutrophil gelatinase associated lipocalin; NSE–neuron specific enolase; sCr–serum creatinine; sNa–serum sodium; Ur–serum urea; TP–Total protein.*
(DOCX)

## Acknowledgments

UK Surgeon General's Department (funding); Dr Amarjit Samra and Yvonne Yau (technical support); Brighton Marathon Medical Research Team, including Dr Rob Galloway and Mrs Carrie Weller (hosting and facilitating); Dr Ed Walter and Dr Oliver Gibson (contribution to data collection).

## Author Contributions

**Conceptualization:** Michael J. Stacey, Neil E. Hill, Iain T. Parsons, Rachael Grimaldi, John P. O'Hara, Stephen J. Brett, David R. Woods.

**Data curation:** Michael J. Stacey, Neil E. Hill, Iain T. Parsons, Rachael Grimaldi, Anna Marshall, Carol House, John P. O'Hara, Stephen J. Brett, David R. Woods.

**Formal analysis:** Michael J. Stacey, Neil E. Hill, Iain T. Parsons, Jenny Wallace, Nishma Shah, David R. Woods.

**Funding acquisition:** Michael J. Stacey, David R. Woods.

**Investigation:** Michael J. Stacey, Neil E. Hill, Iain T. Parsons, Jenny Wallace, Natalie Taylor, Nishma Shah, Anna Marshall, Carol House, David R. Woods.

**Methodology:** Michael J. Stacey, Neil E. Hill, Iain T. Parsons, Rachael Grimaldi, David R. Woods.

**Project administration:** Michael J. Stacey, Neil E. Hill, Iain T. Parsons, Jenny Wallace, Natalie Taylor, Nishma Shah, Carol House.

**Resources:** Michael J. Stacey, Neil E. Hill, Iain T. Parsons, Rachael Grimaldi, Carol House, John P. O'Hara, Stephen J. Brett, David R. Woods.

**Software:** Michael J. Stacey, Neil E. Hill.

**Supervision:** Michael J. Stacey, Neil E. Hill, David R. Woods.

**Writing – original draft:** Michael J. Stacey, Neil E. Hill, Stephen J. Brett, David R. Woods.

**Writing – review & editing:** Michael J. Stacey, Neil E. Hill, Iain T. Parsons, Jenny Wallace, Natalie Taylor, Rachael Grimaldi, Nishma Shah, Anna Marshall, Carol House, John P. O'Hara.

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
