## [Decision Letter · Decision Letter 0]

24 Aug 2021

PONE-D-21-12312

Relative changes in brain and kidney biomarkers with Exertional Heat Illness during a cool weather marathon

PLOS ONE

Dear Dr. Michael J. Stacey,

Thank you for submitting your manuscript to PLOS ONE. After careful consideration, we feel that it has merit but does not fully meet PLOS ONE’s publication criteria as it currently stands. Therefore, we invite you to submit a revised version of the manuscript that addresses the points raised during the review process.

Three expert reviewers with past publications on the topic evaluated the present study. Considering their reports and after reading the manuscript, I believe that the study has merit and presents exciting and novel data. However, despite these positive points, the manuscript can still be improved before it is ready for publication. In particular, additional information is required in the methods section, and findings could be discussed in a biochemical sense (as indicated by the first reviewer).

We look forward to receiving your revised manuscript.

Kind regards,

Samuel Penna Wanner, Ph.D.

Academic Editor

PLOS ONE

Journal Requirements:

3. We noted in your submission details that a portion of your manuscript may have been presented or published elsewhere. “The main manuscript presents entirely new/unpublished data. Related work describing the same cases of EHI, but examining a separate question and biochemical marker, will be uploaded.” Please clarify whether this [conference proceeding or publication] was peer-reviewed and formally published. If this work was previously peer-reviewed and published, in the cover letter please provide the reason that this work does not constitute dual publication and should be included in the current manuscript.

Additional Editor Comments (if provided):

Three expert reviewers with past publications on the topic evaluated the present study. Considering their reports and after reading the manuscript, I believe that the study has merit and presents exciting and novel data. However, despite these positive points, the manuscript can still be improved before it is ready for publication. In particular, additional information is required in the methods section, and findings could be discussed in a biochemical sense (as indicated by the first reviewer).

Please take the three reports into account very seriously. This editor is looking forward to receiving a revised and improved version of the manuscript.

Reviewers' comments:

Reviewer's Responses to Questions

**Comments to the Author**

1. Is the manuscript technically sound, and do the data support the conclusions?

Reviewer #1: Yes

Reviewer #2: Yes

Reviewer #3: Yes

2. Has the statistical analysis been performed appropriately and rigorously? 

Reviewer #1: Yes

Reviewer #2: Yes

Reviewer #3: Yes

3. Have the authors made all data underlying the findings in their manuscript fully available?

Reviewer #1: Yes

Reviewer #2: Yes

Reviewer #3: Yes

4. Is the manuscript presented in an intelligible fashion and written in standard English?

Reviewer #1: Yes

Reviewer #2: Yes

Reviewer #3: Yes

5. Review Comments to the Author

Reviewer #1: Thank you for the opportunity to review the current manuscript. In this work, Stacey and colleagues have examined whether brain-enriched (NSE and S100B) and renal system-related (cysC, KIM-1, and NGAL) biomarkers and copeptin in blood can discriminate marathon runners suffering from EHI and successful marathon finishers without signs of EHI.

The subject of the current investigation is sound and clinically relevant. The literature of biochemical diagnostics of EHI among athletes is scarce. The group have conducted a study that is generally challenging to perform given the potentially methodological caveats such selection bias, incomplete sampling series and small sample size-related issues. It is worth keeping in mind that this study is the first of its kind and the results are therefore preliminary.

I enjoyed reading the paper. My suggestions for improving the manuscript are minor and relate mainly to describing the limitations of the methods, presenting the results, and putting the findings in context in a biochemical sense. Detailed comments below:

Introduction

- Previous literature has been described adequately.

- The clinical problem for which the study is designed is well described.

- The selection of AKI biomarkers seems to be appropriate but for based on the current literature (and also cited publication 16), the decision not to include more brain-specific biomarker GFAP is unfounded.

Methods

- The methods are appropriate overall.

- Studying brain damage with S100B that is known to rise in physical exercise and NSE that is affected by haemolysis without including e.g., GFAP and perhaps UCH-L1 is theoretically unsound. Of course, the strength of the methods is that NSE is a good choice because it is associated with systemic and neural inflammation (e.g., in SAE).

- The main limitation is description of the collapsed runners: i) it is not described whether there were other collapsed runners with EHI in the event than those eight who were enrolled in the study (not possible to assess selection bias, age- and gender matching issues), ii) the demographics of collapsed runners is not described, iii) it is not clear whether collapsed runners had a history of previous kidney conditions, traumatic brain injuries or other CNS conditions such tumours, migraine, stroke of CNS infections. Think that ii) and iii) would have been relatively easy to find out from the runners as they recovered.

Results

- The results section reads well.

- The authors address the decrease of NSE levels in two collapsed runners in the discussion section, but the data is only available in the Supplementary Table 1 and not mentioned in the body text.

- Supplementary Table 1 is labelled incorrectly in the supplementary file, please correct.

Discussion

- The authors put the current findings into context well.

- On the line 320, please correct “radiographically-demonstrable traumatic injury” to “radiographically-demonstrable traumatic intracranial injury”

- In patients with traumatic brain injury, the extent of extracranial confounding in polytrauma patients with TBI has been discussed, but studies suggest that extracranial leak of S100B is quickly eliminated (Savola O, J Trauma. 2004 Jun; 56(6):1229-34; and da Rocha AB. Clin Chem Lab Med. 2006; 44(10):1234-42.). Given the sampling time points are very close to each other, I find very problematic to discuss the S100B levels in collapsed runners.

- The possible demographic differences and also and possible poorer physical performance of collapsed runners compared to successful finishers is a major confounding factor as it is likely has partly exposed them to EHI (reflected in also the biomarker levels). This should be acknowledged.

In summary, this is a well-conducted preliminary study with interesting findings. I am looking forward to seeing the next version of the manuscript.

Reviewer #2: Stacey and colleagues aimed primarily to determine whether some markers could effectively discriminate marathon runners affected by EHI from successful finishers completing the event on the same day. Secondary aims were to examine changes in these biomarkers over time and to relate indirect evidence for organ injury to likely precipitants, co-factors or other indicators relevant to the episodes of collapse observed.

The results are quite relevant. The manuscript is written in a well-organized fashion and has an appropriate language.

Considering that PLOS ONE objectively concentrates on the technical aspects of a study rather than the more subjective evaluations, the presented paper can be published in this journal. However, there are some important questions in the methods to be clarified and corrected.

- The authors say that the time-point of blood collection in the successful finishers and collapsed runners was “as feasible”. It is important you present the real time-point means and standard deviations.

- How was the core temperature measurement? Please insert this information in the methods section.

- Please insert the collapsed group age in table 1 or the text.

- How can the authors argue that the increase in certain blood variables occurred due to the reduction in plasma volume (because of the marathon and dehydration) and not due to real increases in the variables?

- Please insert the relative humidity data and time of the marathon.

- Page 3, line 59, Introduction: The phrase “and can be greater in shorter-distance events” is referenced as number 8. However, the cited article is about “Heat stroke risk for open-water swimmers during long-distance events” and does not support this affirmation.

Reviewer #3: The manuscript etiteled „Relative changes in brain and kidney

biomarkers with Exertional Heat Illness during a cool weather marathon“

was to investigate whether biomarker surrogates for end-organ damage

sampled at point-of-care could discriminate EHI versus successful

marathon performance“ should be accepted in this version because:

-Abstract provided the profile of the manuscript.

-The study gained ethical approval.

-The methods are clear and replicable.

-All the results presented match the methods described and the

statistical analysis appropriate to the research question and study

design.

-The data are presented clearly and appropriately.

-The paper uses appropriate references in the correct style to promote

understanding of the content.

6. PLOS authors have the option to publish the peer review history of their article (what does this mean?). If published, this will include your full peer review and any attached files.

Reviewer #1: **Yes: **Jussi P. Posti

Reviewer #2: **Yes: **Alexandre Sérvulo Ribeiro Hudson

Reviewer #3: No

---

## [Author Response · Author response to Decision Letter 0]

16 Nov 2021

5. Review Comments to the Author

Reviewer #1: Thank you for the opportunity to review the current manuscript. In this work, Stacey and colleagues have examined whether brain-enriched (NSE and S100B) and renal system-related (cysC, KIM-1, and NGAL) biomarkers and copeptin in blood can discriminate marathon runners suffering from EHI and successful marathon finishers without signs of EHI.

The subject of the current investigation is sound and clinically relevant. The literature of biochemical diagnostics of EHI among athletes is scarce. The group have conducted a study that is generally challenging to perform given the potentially methodological caveats such selection bias, incomplete sampling series and small sample size-related issues. It is worth keeping in mind that this study is the first of its kind and the results are therefore preliminary.

I enjoyed reading the paper. My suggestions for improving the manuscript are minor and relate mainly to describing the limitations of the methods, presenting the results, and putting the findings in context in a biochemical sense. Detailed comments below:

Introduction

- Previous literature has been described adequately.

- The clinical problem for which the study is designed is well described.

- The selection of AKI biomarkers seems to be appropriate but for based on the current literature (and also cited publication 16), the decision not to include more brain-specific biomarker GFAP is unfounded.

Many thanks. We have now described the reasons for not incorporating GFAP in this initial preliminary work in our revised discussion, rather than introducing it as a relevant (but un-investigated, in this study) marker at the top of the document, so as not to detract from detail on the markers we do report.

Methods

- The methods are appropriate overall.

- Studying brain damage with S100B that is known to rise in physical exercise and NSE that is affected by haemolysis without including e.g., GFAP and perhaps UCH-L1 is theoretically unsound. Of course, the strength of the methods is that NSE is a good choice because it is associated with systemic and neural inflammation (e.g., in SAE).

Please see above comment on relative dearth of information for SAE prior to the execution of the study, with rationale updated in discussion and future suggestions now including GFP.

- The main limitation is description of the collapsed runners: 

i) it is not described whether there were other collapsed runners with EHI in the event than those eight who were enrolled in the study (not possible to assess selection bias, age- and gender matching issues),

Thank you for assisting us with these clarifying details, we have now categorically stated in the Results that the recruited cases represented 100% of EHI diagnoses diagnosed and initially treated at the on -course medical facilities, while acknowledging in the Discussion that casualties taken from point of collapse on the course direct to hospital may have been missed (though we were aware of no such transfers on this occasion).

 ii) the demographics of collapsed runners is not described, iii) it is not clear whether collapsed runners had a history of previous kidney conditions, traumatic brain injuries or other CNS conditions such tumours, migraine, stroke of CNS infections. Think that ii) and iii) would have been relatively easy to find out from the runners as they recovered.

In practice, once collapsed runners were recovered sufficient to begin providing detailed collateral information on their personal particulars/circumstances, we found that they prioritised self-discharge from medical facilities, as several were visiting from out-of-area and travel arrangements to honour, had friends and family to reunite with, were keen to re-fuel and recover in their own way outside of the medical tents, which were busy with nonEHI presentations. Therefore the demographic data we have provided (gender) represent what was possible for our small teams to collect in the course of processing samples in austere conditions and we do not have retrospective access or general data protection authority to enter the stored clinical notes for more detail on these cases and any potential co-morbidities. 

Results

- The results section reads well.

- The authors address the decrease of NSE levels in two collapsed runners in the discussion section, but the data is only available in the Supplementary Table 1 and not mentioned in the body text.

Now addressed in main body text, thank you.

- Supplementary Table 1 is labelled incorrectly in the supplementary file, please correct.

Corrected in line with preferred journal formatting, thank you.

Discussion

- The authors put the current findings into context well.

- On the line 320, please correct “radiographically-demonstrable traumatic injury” to “radiographically-demonstrable traumatic intracranial injury”

We have done so, thank you

- In patients with traumatic brain injury, the extent of extracranial confounding in polytrauma patients with TBI has been discussed, but studies suggest that extracranial leak of S100B is quickly eliminated (Savola O, J Trauma. 2004 Jun; 56(6):1229-34; and da Rocha AB. Clin Chem Lab Med. 2006; 44(10):1234-42.). Given the sampling time points are very close to each other, I find very problematic to discuss the S100B levels in collapsed runners.

Unfortunately we could not find supporting data on rates of in vivo S100B decay in the paper by Savalo et al (kindly referenced by the reviewer), but did note that sampling was completed within 6 hours in the various degrees of injury assessed and showed discrimination for brain injury and large extracranial injury versus smaller extracranial injuries and control subjects. In the other paper suggested, by da Rocha et al, critically brain-injured patients were sampled at study entry T0 (median 10.9 h post injury), T+1 day and T+7 days, showing significant relative elevations for T0 vs T24 and between survivors/non survivors at T24. While we acknowledge that, for good practical reasons, studies recruiting brain injured patients from trauma are limited in their ability to assay S100B at close to point-of-incapacity as we were able to do for EHI cases in the marathon – and this leaves a relative gap in that literature – data do exist to demonstrate cellular release of S100B as early as 3 hours post-injury in vitro. Though limited to 3 cases with data beyond the initial 30 minute period (T0) following incapacitation, our S1 supplementary table does show mild to moderate reductions in S100B at 1 hour post-insult (T1) – possibly compatible with correction of physiological haemoconcentration post-run and fluid resuscitation/recovery – but also, importantly, a relative rise at 4 hours in the one EHI case unwell enough to be admitted from the course facilities to the local hospital. Therefore if our study had been able to recruit and retain greater numbers of EHI cases with relatively greater severity of illness/injury over serial sampling points, as we intended, S100B may have presented greater interest/potential utility, especially at this later timepoint. As things stand, we feel that the S100B data nicely complement the (more neurally-specific) NSE data and that novelty remains in this comparison overall. Nevertheless, if the Editor feels strongly that S100B should be removed from the manuscript for publication, we would strongly consider this.

- The possible demographic differences and also and possible poorer physical performance of collapsed runners compared to successful finishers is a major confounding factor as it is likely has partly exposed them to EHI (reflected in also the biomarker levels). This should be acknowledged.

Respectfully, we tend to see things the other way, and would rather acknowledge the possibility that runners who experienced EHI were more likely to be highly motivated, as has been reported in post-EHI interviewing of military cases (ABRIAT, A., BROSSET, C., BRÉGIGEON, M. & SAGUI, E. 2014. Report of 182 cases of exertional heatstroke in the French Armed Forces. Military medicine, 179, 309-314), and perhaps ‘fit enough’ to drive themselves to the point of collapse. On balance, we would like to acknowledge the need for a future study with better potential to characterise collapsed cases, including in-race monitoring/surveillance (as can be achieved with the chips issued with running numbers as an increasingly ubiquitous standard for runners to log and retrospectively review their time ‘splits’ across the distance) and post-event questionnaires to including training volumes, predicted race times etc, as well as a bigger team to capture the more basic demographic data highlighted above.

In summary, this is a well-conducted preliminary study with interesting findings. I am looking forward to seeing the next version of the manuscript.

We are very grateful for your time, input and expertise, thank you.

Reviewer #2: Stacey and colleagues aimed primarily to determine whether some markers could effectively discriminate marathon runners affected by EHI from successful finishers completing the event on the same day. Secondary aims were to examine changes in these biomarkers over time and to relate indirect evidence for organ injury to likely precipitants, co-factors or other indicators relevant to the episodes of collapse observed.

The results are quite relevant. The manuscript is written in a well-organized fashion and has an appropriate language.

Considering that PLOS ONE objectively concentrates on the technical aspects of a study rather than the more subjective evaluations, the presented paper can be published in this journal. However, there are some important questions in the methods to be clarified and corrected.

- The authors say that the time-point of blood collection in the successful finishers and collapsed runners was “as feasible”. It is important you present the real time-point means and standard deviations.

We agree that real time-pint means would be ideal. However the conditions prevailing in this field based study of mass participation exercise - with emergency responders recovering collapsed casualties from the course to the appropriate medical facilities and conducting clinical handover under pressure of time in busy medical tents – meant that in a number of the casualties timings were provided as 10 or 20 minute intergers, with sampling performed at a similarly approximate time point, but confirmed as no more than 30 minutes from collapse. Even the intensive care environments from which the neuro-biomarkers have been reported in brain injury and sepsis-associated encephalopathy are relatively controlled, but studies often default to the nearest whole hour post-injury in reporting. We have previously published field work in which copeptin was sampled within 20 minutes of exertion ceasing (Stacey MJ, Delves SK, Britland SE,et al. Eur J Appl Physiol. 2018;118:75-84.) and deemed 30 minutes to be acceptable in this instance, given the added complications of marathon logistics and, ultimately, the practical question we were attempting to address of how these markers might identify/discriminate EHI. We feel that the data support this approach, with high discriminant value shown by both NSE and copeptin. 

- How was the core temperature measurement? Please insert this information in the methods section.

In the Methods section on collapsed runners, we have already stated ‘core body temperature measured rectally (Intellivue integrated thermistor, Philips Healthcare, Amsterdam, Netherlands).’ Please do let us know if further clarification is required.

- Please insert the collapsed group age in table 1 or the text.

Please see comments above about the logistical constraints that limited the fuller data collection that would otherwise have been ideal in this situation. While we do have date of birth data for the casualties treated at Medical Facility I, Medical Facility II (finish-line) was dealing with high volumes of non-EHI (including hypothermia, myocardial infarction and post-exercise collapse, some of which was precipitated by the relatively inclement conditions) and introduced limitations to data collection beyond confirming the EHI diagnosis has been assigned clinically, the gender of the patient and their informed consent to participate in the study. A larger team may have improved the possibility of adding further casualty data, however we were necessarily split across two sites which limited availability of personnel and was in fact necessary given that half of the cases presented at Medical Facility I.

- How can the authors argue that the increase in certain blood variables occurred due to the reduction in plasma volume (because of the marathon and dehydration) and not due to real increases in the variables?

We apologise if we have not been clear here – in the statement on copeptin, the stable surrogate for arginine vasopressin, we have now noted that ‘both hypovolaemia and hypertonicity being potent, classical stimuli to the release of AVP/copeptin.’

- Please insert the relative humidity data and time of the marathon.

We have now inserted the hours of the event, as suggested, in relation to relevant meteorological conditions in the first line of the Results. We do not have access to raw relative humidity data, however, but do have WBGT recordings for the hours in which the marathon was conducted. However our understanding is that relative humidity and wet bulb measures have little relevance to thermoregulation in marathon conditions when the ambient temperature is as low as we report for the study. Therefore WBGT has been omitted; we could provide this as a supplementary data table, but have been informed previously in relation to another study that this would be nonsensical at low ambient temperature.

.

- Page 3, line 59, Introduction: The phrase “and can be greater in shorter-distance events” is referenced as number 8. However, the cited article is about “Heat stroke risk for open-water swimmers during long-distance events” and does not support this affirmation.

Thank you, our apologies, for brevity we have removed the statement and the duplicate reference (number 9 appearing as both 8 and 9, our error) is now singular and appropriately positioned.

Reviewer #3: The manuscript etiteled „Relative changes in brain and kidney

biomarkers with Exertional Heat Illness during a cool weather marathon“

was to investigate whether biomarker surrogates for end-organ damage

sampled at point-of-care could discriminate EHI versus successful

marathon performance“ should be accepted in this version because:

-Abstract provided the profile of the manuscript.

-The study gained ethical approval.

-The methods are clear and replicable.

-All the results presented match the methods described and the

statistical analysis appropriate to the research question and study

design.

-The data are presented clearly and appropriately.

-The paper uses appropriate references in the correct style to promote

understanding of the content.

Thank you.

---

## [Decision Letter · Decision Letter 1]

26 Jan 2022

PONE-D-21-12312R1Relative changes in brain and kidney biomarkers with Exertional Heat Illness during a cool weather marathonPLOS ONE

Dear Dr. Michael J. Stacey,

Thank you for submitting your manuscript to PLOS ONE. After careful consideration, we feel that it has merit but does not fully meet PLOS ONE’s publication criteria as it currently stands. Therefore, we invite you to submit a revised version of the manuscript that addresses the points raised during the review process. After reading the manuscript and taking the two reviewers' comments into account, I believe that the revised manuscript was improved compared to the previous version submitted to PLOS One. Indeed, the manuscript is close to being ready for publication. The authors were highly responsive to all comments made by the first reviewer, but they were not to those by the second reviewer. Please see the editor's comments at the end of this letter.

We look forward to receiving your revised manuscript.

Kind regards,

Samuel Penna Wanner, Ph.D.

Academic Editor

PLOS ONE

Additional Editor Comments:

1) If the authors answer the second reviewer adequately, I will accept the manuscript without another round of external reviews.

2) If possible, please provide information on the age of runners experiencing exertional heat illness. If not possible, the Academic Editor understands the unfavorable/unpredictable conditions of data collection.

3) Please consider that the lack of information on the exercise-induced change in plasma volume (as determined from hematocrit and hemoglobin measurements) is a limitation of the present study. Please also consider carrying out these analyses in your future studies because they may be academically relevant.

4) Please indicate the volume of blood samples.

5) The authors wrote the following information twice in the manuscript: “henceforth referred to as ‘successful finishers’” (lines 118 and 119, 140 and 141). The authors may want to delete the repeated information in lines 140 and 141.

6) Line 144. Please consider replacing “mins” with “minutes”. It seems that the authors have not abbreviated this measuring unit throughout the manuscript.

7) Line 222. Please consider replacing “14:00 and 15:00 PM” with “14:00 and 15:00” or “2:00 and 3:00 PM”.

8) Please consider providing updated information for references 7 and 14.

9) Please consider providing complete information for reference 29 (i.e., the volume, issue, and page numbers).

10) Please consider reducing the excessive white space between panels in Figure 1.

11) Please consider removing a blank column in the supplementary Table 1.

12) The Academic Editor agrees with the maintenance of data regarding S100B.

Reviewers' comments:

Reviewer's Responses to Questions

**Comments to the Author**

1. If the authors have adequately addressed your comments raised in a previous round of review and you feel that this manuscript is now acceptable for publication, you may indicate that here to bypass the “Comments to the Author” section, enter your conflict of interest statement in the “Confidential to Editor” section, and submit your "Accept" recommendation.

Reviewer #1: All comments have been addressed

Reviewer #2: (No Response)

2. Is the manuscript technically sound, and do the data support the conclusions?

Reviewer #1: Yes

Reviewer #2: Partly

3. Has the statistical analysis been performed appropriately and rigorously? 

Reviewer #1: Yes

Reviewer #2: Yes

4. Have the authors made all data underlying the findings in their manuscript fully available?

Reviewer #1: Yes

Reviewer #2: Yes

5. Is the manuscript presented in an intelligible fashion and written in standard English?

Reviewer #1: Yes

Reviewer #2: Yes

6. Review Comments to the Author

Reviewer #1: Thank you for modifying the manuscript as per my comments. I am happy with the paper as it stands now.

(this review was already submitted on 6th December, but it seems that there has been somekind of an error)

Reviewer #2: I am grateful to the authors for considering my review. However, I emphasize that two major questions were not clarified at all.

The authors conclude that “The novel biomarker candidates for EHI outcomes in cool-weather endurance exercise, early elevations in NSE and copeptin provided sufficient discrimination to suggest utility at point-of-incapacity” however the two major points I indicate below should be answered, thus the conclusion of the article will be more honest.

First, it is essential to indicate more characteristics of the sample, especially those affected with EHI. For example, the interpretation of the data would change if all 8 volunteers were over 50 years old. The present reviewer does not understand how such simple information can be neglected in the study, as the authors themselves claimed to have had some kind of contact with the sample after they recovered: “For EHI cases who initially lacked mental capacity to consent for themselves ( AVPU grade V to U, or A with any concern for capacity) we proceeded with presumed consent until they were deemed able to give it retrospectively.” In addition to age, another variable as BMI would be of great importance.

Second, with my question “How can the authors argue that the increase in certain blood variables occurred due to the reduction in plasma volume (because of the marathon and dehydration) and not due to real increases in the variables?” I was expecting the authors to argue whether they corrected the values of the biomarkers for variations in plasma volume (DILL, DB; COSTILL, DL Calculation of percentage changes in volumes of blood, plasma, and red cells in dehydration. J Appl Physiol, v. 37, No. 2, pp. 247-248, 1974). If this correction was not made, I suggest putting this limitation in the study.

Lastly a minor question: Which was the volume of blood samples collected on the time-points?

These simple information requested would be of great relevance to the researchers in this field, as this is an article with unprecedented results.

7. PLOS authors have the option to publish the peer review history of their article (what does this mean?). If published, this will include your full peer review and any attached files.

Reviewer #1: **Yes: **Jussi P. Posti

Reviewer #2: **Yes: **Alexandre Sérvulo Ribeiro Hudson

---

## [Author Response · Author response to Decision Letter 1]

27 Jan 2022

Additional Editor Comments: Reponses

1) If the authors answer the second reviewer adequately, I will accept the manuscript without another round of external reviews.

Thank you, we have endeavoured to do so below.

2) If possible, please provide information on the age of runners experiencing exertional heat illness. If not possible, the Academic Editor understands the unfavorable/unpredictable conditions of data collection.

Thank you for your understanding, please see below.

3) Please consider that the lack of information on the exercise-induced change in plasma volume (as determined from hematocrit and hemoglobin measurements) is a limitation of the present study. Please also consider carrying out these analyses in your future studies because they may be academically relevant.

We shall, please see full response below.

4) Please indicate the volume of blood samples.

13 ml blood, please see below.

5) The authors wrote the following information twice in the manuscript: “henceforth referred to as ‘successful finishers’” (lines 118 and 119, 140 and 141). The authors may want to delete the repeated information in lines 140 and 141.

Thank you, we have done so.

6) Line 144. Please consider replacing “mins” with “minutes”. It seems that the authors have not abbreviated this measuring unit throughout the manuscript.

Thank you, we have done so.

7) Line 222. Please consider replacing “14:00 and 15:00 PM” with “14:00 and 15:00” or “2:00 and 3:00 PM”.

Thank you, we have done so.

8) Please consider providing updated information for references 7 and 14.

Thank you, we have done so.

9) Please consider providing complete information for reference 29 (i.e., the volume, issue, and page numbers).

We have done so for reference 28, which we assume you meant – apologies for this omission.

10) Please consider reducing the excessive white space between panels in Figure 1.

11) Please consider removing a blank column in the supplementary Table 1.

Thank you for spotting this, we have done so.

12) The Academic Editor agrees with the maintenance of data regarding S100B.

Thank you.

Reviewer 2 Comments: Reponses

The authors conclude that “The novel biomarker candidates for EHI outcomes in cool-weather endurance exercise, early elevations in NSE and copeptin provided sufficient discrimination to suggest utility at point-of-incapacity” however the two major points I indicate below should be answered, thus the conclusion of the article will be more honest.

First, it is essential to indicate more characteristics of the sample, especially those affected with EHI. For example, the interpretation of the data would change if all 8 volunteers were over 50 years old. The present reviewer does not understand how such simple information can be neglected in the study, as the authors themselves claimed to have had some kind of contact with the sample after they recovered: “For EHI cases who initially lacked mental capacity to consent for themselves ( AVPU grade V to U, or A with any concern for capacity) we proceeded with presumed consent until they were deemed able to give it retrospectively.” In addition to age, another variable as BMI would be of great importance.

We have been unable to do so – the nature of a recreational event seems to favour participants who have been incapacitated wanting to move swiftly on with their day/conclusion to the unfortunate event of EHI supervening, hence conditions of data collection were suboptimal. We have accounted for this and the limitation below at lines 401-407. 

Second, with my question “How can the authors argue that the increase in certain blood variables occurred due to the reduction in plasma volume (because of the marathon and dehydration) and not due to real increases in the variables?” I was expecting the authors to argue whether they corrected the values of the biomarkers for variations in plasma volume (DILL, DB; COSTILL, DL Calculation of percentage changes in volumes of blood, plasma, and red cells in dehydration. J Appl Physiol, v. 37, No. 2, pp. 247-248, 1974). If this correction was not made, I suggest putting this limitation in the study.

Please see lines 401-407.

Lastly a minor question: Which was the volume of blood samples collected on the time-points?

These simple information requested would be of great relevance to the researchers in this field, as this is an article with unprecedented results.

4.5 ml for plasma, 8.5 ml for serum = 13 ml, now documented in Methods section at lines 140 and 169.

---

## [Editor Report · Decision Letter 2]

31 Jan 2022

Relative changes in brain and kidney biomarkers with Exertional Heat Illness during a cool weather marathon

PONE-D-21-12312R2

Dear Dr. Michael J. Stacey,

We’re pleased to inform you that your manuscript has been judged scientifically suitable for publication and will be formally accepted for publication once it meets all outstanding technical requirements.

Kind regards,

Samuel Penna Wanner, Ph.D.

Academic Editor

PLOS ONE

Additional Editor Comments (optional):

After reading the revised manuscript, I believe that the authors have adequately addressed all the minor points the second reviewer and I (i.e., the Academic editor) have raised. Thank you! The manuscript deserves to be published in PLOS One in its current form. Congratulations. I am looking forward to seeing your future study on this topic.
---

## [Editor Report · Acceptance letter]

7 Feb 2022

PONE-D-21-12312R2 

Relative changes in brain and kidney biomarkers with Exertional Heat Illness during a cool weather marathon 

Dear Dr. Stacey:

I'm pleased to inform you that your manuscript has been deemed suitable for publication in PLOS ONE. Congratulations! Your manuscript is now with our production department. 

Kind regards, 

on behalf of

Dr. Samuel Penna Wanner 

Academic Editor

PLOS ONE